# Contribution of Plantar Fascia and Intrinsic Foot Muscles in a Single-Leg Drop Landing and Repetitive Rebound Jumps: An Ultrasound-Based Study

**DOI:** 10.3390/ijerph18094511

**Published:** 2021-04-23

**Authors:** Masanori Morikawa, Noriaki Maeda, Makoto Komiya, Arisu Hirota, Rami Mizuta, Toshiki Kobayashi, Kazuki Kaneda, Yuichi Nishikawa, Yukio Urabe

**Affiliations:** 1Division of Sport Rehabilitation, Graduate School of Biomedical and Health Sciences, Hiroshima University, 1-2-3 Kasumi, Minami-ku, Hiroshima 734-8553, Japan; m-masanori@hiroshima-u.ac.jp (M.M.); norimmi@hiroshima-u.ac.jp (N.M.); makoto-komiya@hiroshima-u.ac.jp (M.K.); arisu-hirota@hiroshima-u.ac.jp (A.H.); m202263@hiroshima-u.ac.jp (R.M.); kazuki-kaneda@hiroshima-u.ac.jp (K.K.); 2Department of Biomedical Engineering, Faculty of Engineering, The Hong Kong Polytechnic University, 11 Yuk Choi Road, Hung Hom, Hong Kong, China; toshiki.kobayashi@polyu.edu.hk; 3Faculty of Frontier Engineering, Institute of Science & Engineering, Kanazawa University, Kanazawa 076-264-5111, Japan; yuichi@se.kanazawa-u.ac.jp

**Keywords:** muscle thickness, cross-sectional area, muscle hardness, drop landing, reactive jump index

## Abstract

The plantar fascia and intrinsic foot muscles (IFM) modulate foot stiffness. However, it is unclear whether the corresponding ultrasonography findings reflect it. This study aimed to examine the effect of the plantar fascia and IFM morphologies on force attenuation during landing and reactivity when jumping in healthy adults (*n* = 21; age, 21–27 years). Thickness, cross-sectional area (CSA), and hardness of the plantar fascia, abductor hallucis (AbH), and flexor hallucis brevis (FHB) muscles were measured using ultrasonography. Single-leg drop landing and repetitive rebound jumping tests assessed the ground reaction force (GRF) and reactive jump index (RJI), respectively. The CSA of FHB was negatively correlated with maximum vertical GRF (r = −0.472, *p* = 0.031) in the single-leg drop landing test. The CSA of AbH was negatively correlated with contact time (r = −0.478, *p* = 0.028), and the plantar fascia thickness was positively correlated with jump height (r = 0.615, *p* = 0.003) and RJI (r = 0.645, *p* = 0.002) in the repetitive bound jump test. In multivariate regression analysis, only the plantar fascia thickness was associated with RJI (β = 0.152, 95% confidence interval: 7.219–38.743, *p* = 0.007). The CSA of FHB may contribute to force attenuation during landing. The thickness of the plantar fascia and CSA of AbH may facilitate jumping high with minimal contact time.

## 1. Introduction

The human foot comprises 28 bones that form three flexible arch structures. The stiffness of these foot arches is passively and actively modulated by the plantar fascia and the intrinsic foot muscles (IFM), respectively [1]. The plantar fascia and IFM decrease or increase foot stiffness to attenuate external forces or transmit the internal force of the extrinsic foot muscle to the foot, respectively [2]. The IFM have been shown to be activated according to load-induced arch deformations [3]. Moreover, IFM can protect the plantar fascia from excessive strain or modulate foot stiffness in response to increased postural and loading demands [4].

In sports activities, force attenuation while landing is necessary to prevent injury [5]; meanwhile, lower-limb stiffness at the time of jumping contributes to the ability to jump higher with minimal ground contact time [6]. Indeed, athletes land and jump during training and competition [7], acting on the plantar fascia and IFM to attenuate and activate different types of force, as required.

In recent years, ultrasonography has been used to evaluate the morphologies of the plantar fascia and IFM; their relationship with performance has been investigated in gait [8], running [9], and static [10] and dynamic [11] balance tests. However, it remains unclear whether the morphologies of the plantar fascia and IFM are related to force attenuation during landing or reactivity during jumping. This study aimed to examine the relationship between the plantar fascia and IFM morphologies, and force attenuation during landing and reactivity when jumping among healthy adults.

## 2. Materials and Methods

Twenty-three healthy recreationally active collegiate students were recruited for participation in this study from Hiroshima University. “Recreationally active” was defined as participation in at least 150 min of moderate activity per week for at least 6 months [12]. Individuals were ineligible for this study if they met any of the following criteria: (1) history of lower limb trauma; (2) neurological conditions; and (3) plantar fasciitis, ligamentous injuries, bursitis, or orthopedic injuries to the lower limb.

The study was conducted in accordance with the principles of the Declaration of Helsinki and was approved by the Ethics Committee for Epidemiology of Hiroshima University (approval number: E-2090). All participants provided informed consent to study participation.

### 2.1. B-Mode Ultrasonography and Real-Time Tissue Elastography

Morphologies of IFM and plantar fascia (thickness, cross-sectional area (CSA), and hardness) were imaged by B-mode ultrasonography (HI VISION Avius; Hitachi Aloka Medical, Tokyo, Japan) and real-time tissue elastography (RTE). An 8-MHz linear array probe was used to obtain a continuous ultrasound image. An acoustic coupler (EZU-TECPL1; Hitachi Aloka Medical, Tokyo, Japan) was put on the transducer with a plastic attachment (EZU-TEATC1; Hitachi Aloka Medical, Tokyo, Japan) and used as reference material. The hardness value of reference material was 22.6 ± 2.2 kPa, which was a relative value based on the acoustic coupler in the material tests conducted by the manufacturer.

### 2.2. Evaluation of Thickness and CSA

Thickness and CSA values of selected tissue were measured with B-mode ultrasonography, as reported previously [10,13]; the aim was to measure the selected tissue in the contraction-free position using a linear array probe. The following measurements were taken in the prone position, with the subjects’ ankles in a neutral position and knees flexed at 90° to minimize pressure on the tissues while ensuring that the probe made the best contact with the skin:Thickness of the abductor hallucis (AbH), flexor hallucis brevis (FHB), and flexor digitorum brevis (FDB) muscles;CSA of AbH, FHB, and FDB, and;Thickness of the plantar fascia in the calcaneus.

The position of the probe was marked on the skin surface with semipermanent ink. The probe was placed along a line perpendicular to the long axis of the foot on the anterior aspect of the medial malleolus to record the CSA image of AbH; it was subsequently placed approximately perpendicular to the same line to record the thickness image. Furthermore, the probe was placed perpendicular to a line parallel to FHB to record its CSA image and was subsequently placed along the same line to record the thickness image. Moreover, the probe was placed along a line between the third toe and the medial calcaneal tubercle to record the CSA image of the FDB muscle and was subsequently placed along the same line to record the thickness image. Moreover, the probe was placed along a line between the second toe and the medial calcaneal tubercle to record the thickness image (Figure 1). These measurements are reliable in assessing the plantar muscle and plantar fascia thickness and CSA [13,14].

### 2.3. Evaluation of IFM and Plantar Fascia Hardness

The head of the probe was coated with a transmission gel to achieve acoustic coupling. The RTE was imaged by manually applying repetitive light compression (rhythmic compression–relaxation cycle) with the transducer in the scan position. To obtain appropriate images for investigation, the examiner responsible for ultrasonography assessment pressed the transducer against the selected tissue with constant repeated pressure (the strain ratio in the range of −0.7 to 0.7) and monitored the pressure indicator incorporated into the ultrasound scanner (Figure 2). For each task, three images were randomly selected, and the average values were calculated after scanning. The hardness and thickness of the AbH, FHB, and FDB muscles and plantar fascia were measured in the supine position, with the subjects’ ankles in neutral position and their knees flexed at 90°. The RTE assessment took less than 10 min. The RTE images were displayed as translucent, color-coded, real-time images superimposed on the B-mode images, with the red signal indicating a large distortion in response to the compression (the softest component), blue indicating the least distortion (the hardest component), and green representing the mean strain.

The mean strain in the region of interest (ROI) was monitored on a strain graph to adjust the force and frequency of compression. When the elastogram was successfully superimposed on the ROI, the frequency was adjusted from 2 Hz to 4 Hz to maximize the image quality. The strain rate within each ROI was measured automatically, using built-in software, and the strain ratio (tissue/reference ratio; the value of the strain measured in the reference ROI divided by the corresponding value obtained in the tissue ROI) was calculated per image. The RTE in response to the compression force is smaller in harder than in softer tissue; hence, as the tissues become stiffer, the RTE value becomes smaller [15]. RTE measurements were performed three times, and the mean value within the ROI from these measurements was used in subsequent analyses. All measurements and positioning of all ROIs were performed by the same examiner. High intra-observer reproducibility was previously reported for the tissue/reference ratio of RTE measurements (tissue/coupler) [16,17].

### 2.4. Single-Leg Drop Landing Test

The single-leg drop landing test was conducted using the center of pressure (COP) trajectory, and the vertical ground reaction force was measured using a force plate (Type 9281 B; Kistler, Switzerland), 10 cm in height. The participants stood one-legged on their right foot, static and barefoot on a box, 30 cm in height. They were asked to jump in a forward direction from the box on a cue given by voice guidance from a personal computer with relevant software (Dynamic balance evaluation applications ver. 1.2.5.1, Technology Service Co., Ltd., Nagano, Japan). They were instructed to jump with their hands folded in front of their chest, landing with their right foot on the center of the force plate placed in front of the box. The participants were asked to maintain the single-legged standing posture for 5 s after landing, and to continue keeping both hands folded in the same position. A 10-s interval was set between trials. Trials were excluded from the data analysis when a participant (1) could not maintain a single-leg standing posture for 5 s after landing; (2) was disengaged from the force plate during or after landing; (3) moved the base of support by shifting their foot after landing, trying to maintain standing posture; or (4) released their folded arms. The landing test was repeated until three successful trials were performed and used for data analysis.

### 2.5. Repetitive Rebound Jump Test

Repetitive rebound jump (RJ) performance was assessed using an Optojump^TM^ system (Microgate, Bolzano, Italy). This system consists of two infrared photocell bars, with one bar acting as a transmitter unit containing 96 light-emitting diodes positioned 0.003 m above the ground, while the other bar acts as a receiver unit. Participants were instructed to keep their hands on their hips to avoid upper-body interference, while jumping and landing repetitively. They were asked to land on the same spot each time with legs initially extended and then flexed, while looking ahead at all times. Additionally, the participants were asked to jump as high as possible and minimize the ground contact time, while jumping again immediately. When a participant performed the RJ within the parallel bars placed on the ground, the light was interrupted by the participant’s foot during the jump, which triggered the sensor in the unit to record a sampling frequency of 1000 Hz. Two sets of seven repetitive rebound jumps were performed with a 5-min rest between sets. Of seven repetitive jumps in the second set, the second through fifth jumps were used in the analysis, as this approach is associated with good inter-day reliability [18].

### 2.6. Data Collection

The voltage signal output from the force plate was amplified by an amplifier (Type9865E1Y28, Kistler, Switzerland) and digitized by an A/D converter (NI-USB6218BNC, National Instruments, Austin, Texas, USA) at a sampling frequency of 1000 Hz. The force plate data were filtered using a zero-lag, fourth-order, low-pass Butterworth filter with a frequency cut-off of 15 Hz. The vertical ground reaction force was analyzed for 5 s after ground contact and was defined as a vertical component of >5 N. The locus length of the center of pressure (COP) normalized by foot length (% foot length) was calculated to quantify the magnitude of postural sway and the maximum vertical ground reaction force normalized by body weight (VGRFmax) (% body weight, BW); time to peak (ms) and loading rate (% BW/ms) were similarly estimated to quantify force attenuation during landing. The loading rate was defined as the peak of the first derivative of the force–time curve [19]. The average of the values obtained in three successful trials was used for the additional analysis.

Optojump^TM^ proprietary software (OptojumpTM Next software, version 1.9.9.0) was used to automatically calculate the RJ performance variable, which included jump height (cm), contact time (s), and reactive jump index (RJI) (m/s). The jump height was estimated as:(1/2 × Tair × g)^2^ × (2 g)^−1^,(1)
where Tair is the flight time (s) from the force record on the force plate, and g is the acceleration due to gravity (9.81 m/s^2^). The RJI was estimated as 1/8 × g × Tair^2^/contact time. The average of the values obtained from five repetitive jumps was used for the additional analysis. Rebound jump index was used for assessing reactive strength, which is an athlete’s ability to efficiently brake and absorb (eccentric) forces within specific time frames, before subsequently generating a propulsive (concentric) force [18].

### 2.7. Statistical Analysis

Data were analyzed using SPSS (IBM Japan Co., Ltd., Tokyo, Japan). The Shapiro–Wilk test was used to test the normality of data distribution. Normally and non-normally distributed variables were presented as the mean ± standard deviation (SD) and the median and interquartile range (IQR), respectively. Correlation coefficients were calculated with the Pearson product-moment correlation test or Spearman rank correlation test, depending on data distribution, and were used to estimate the possible relationships between thickness, CSA, and RTE values of the assessed tissue (AbH, FDB, and FHB muscles and plantar fascia), and rebound jump and drop landing assessment results. *p*-values of < 0.05 were considered statistically significant.

Multiple linear regression with forced entry was performed to examine the association between selected tissue morphology and rebound jump performance, which was used as an RJI. To avoid overfitting due to the small sample size, we used variables with correlation coefficients of <0.10 between selected tissue morphology and RJ performance as objective variables.

As performed in a previous study [20], a post hoc power analysis was conducted. The procedure accounted for the population effect size (f^2^), the “α err prob,” sample size, and the number of predictors included in the regression model to estimate the power of the omnibus *F*-test (“Power [1 err prob]”).

## 3. Results

Two of 23 prospective participants were excluded from this study due to hallux valgus. The mean, median, IQR, and SD values of ultrasonographic evaluations and repetitive rebound jump and drop landing assessments are presented in Table 1.

The CSA of FHB was negatively correlated with VGRFmax in forward drop landing (r = −0.472, *p* = 0.031) (Table 2). The thickness of the plantar fascia was negatively correlated with the COP length (r = −0.513, *p* = 0.018). In addition, the AbH muscle size and plantar fascia thickness were correlated with increased RJ performance.

Table 3 summarizes the results of multiple regression analysis of the association between IFM and plantar fascia morphology and RJ performance. There was no association between contact time, the CSA of AbH, and the thickness of the plantar fascia. Only the thickness of the plantar fascia was associated with the RJI. In post hoc power analysis, the multivariate regression model for the association between ultrasonographic assessment items and the rebound jump index demonstrated sufficient power (1 err prob = 0.836).

## 4. Discussion

The main finding of this study indicated that the morphology of the FHB muscle and plantar fascia was related to force attenuation in single-leg drop landing; meanwhile, the AbH muscle and the plantar fascia were related to reactivity in the repetitive rebound jump test. Moreover, the thickness of the plantar fascia is associated with the RJI. Since the RJI is considered to be the indicator of an athlete’s ability to efficiently brake and absorb eccentric forces within specific time frames, before subsequently generating a propulsive concentric force, this present study suggests that the thicker plantar fascia may indicate an ability to jump higher with minimal contact time during jumping.

Olsen et al. [21] reported the kinematic characteristics of the metatarsophalangeal (MTP) joint during forward drop landing. At the moment of ground contact, the first MTP joint was rapidly extended to 20°. This phenomenon may increase the absorptive function of the foot by engaging the windlass mechanism and tightening the plantar fascia. In this study, we showed that a larger FHB correlated with a lower VGRFmax during single-leg drop landing. This finding suggests that the eccentric contraction of the FHB muscle may attenuate the force in response to toe extension during ground contact. Another possible explanation is that the FHB muscle is anchored to the calcaneus in the medial tubercle region (i.e., the origin of the plantar fascia) through its attachment to the medial intermuscular septum [22]. In other words, as in the windlass mechanism of the plantar fascia, the contraction of the FHB muscle may modulate the stiffness of the MTP joint and foot.

The present study showed a significant negative correlation between plantar fascia thickness and COP length in single-leg drop landing. Generally, COP is transferred by postural sway. If its structure is too flexible, postural sway would increase because it needs to be compensated by proximal joint. In other words, the ability to regulate the stiffness of the foot structure is considered to be important for reducing COP displacement. The plantar fascia is a fiber that attaches from the calcaneus to the proximal phalanx, and can change the stiffness of the foot structure through the windlass mechanism [23]. Thus, in the case of single-leg drop landing phase, the plantar fascia would be stretched by the extension of the toes and contraction of the triceps surae muscle, and can respond to the postural demand for loading. A thick plantar fascia may have a high potential for that demand, thus the correlation between plantar fascia thickness and COP may have been observed in the current study.

To the best of our knowledge, this is the first study to showing a relationship between the CSA of AbH and RJI. A previous study showed that the RJI is positively correlated with the height of the medial longitudinal arch [24]. This finding indicates that the medial longitudinal arch plays a role in storing and releasing elastic strain energy during repetitive dynamic movement. AbH is the largest single muscle of the IFM [25] and arises from the process of calcaneal tuberosity at the base of the first phalanx of the hallux and supports the medial longitudinal arch. Indeed, the CSA of AbH contributes to the height of the medial longitudinal arch [26]. The present findings expand on the relationship between the RJI and the IFM, using ultrasonography assessment.

This study has demonstrated an association between the thicker plantar fascia and greater RJI. A previous study explored the relationship between the thickness of the plantar fascia and the RJI and reported likely correlation between them [27]. The RJI reflects the ability to jump as high as possible with minimal contact time; the spring-like function of the foot is important to RJ because it efficiently attenuates the impact in the landing phase and generates explosive power in the take-off phase. In particular, the plantar fascia contributes to attenuating the impact and managing the stiffness of the foot through the bow-string arrangement of the medial longitudinal arch [28]. Moreover, as the triceps fascia tension increases, so does that of the plantar fascia [1]. This is called the Achilles–calcaneus–plantar system, which is known to effectively transport the force generated by the lower limb to the foot during a jump [29]. Overall, the thicker the plantar fascia, the higher the potential for storing and releasing energy, which may improve RJ performance in healthy adults.

Previously, it was unclear if the morphology assessed by ultrasound of the plantar fascia and IFM could reflect sports performance during the drop landing and repetitive rebound jump. In the present study, these soft tissue morphologies and actual sports performance were correlated. Therefore, this indicated the possibility to assess the plantar fascia and IFM by using ultrasound imaging devices in order to quantify sports performance in the future. In addition, it was also unknown which of plantar fascia or IFM affect attenuating force during landing and repetitive rebound jumping. This study demonstrated the relationship between the plantar fascia, flexor hallucis brevis, and abductor pollicis brevis and sports performances. These results may be important findings for specific training for the foot and consideration of the mechanism of injury in the future.

This study has some limitations that need to be considered when interpreting its findings. First, this study only included healthy adults without foot abnormalities. However, previous studies have shown that the plantar fascia is thicker in populations with plantar tendinitis [30]; thus, a relationship different from that presented in this study may be observed in this population. Second, this study did not account for the impact of sex. A previous study reported that the menstrual cycle may affect the thickness of the plantar fascia. However, this association was not examined in this study; therefore, further research is required to elucidate the differences between men and women in this context. Third, we assessed the foot morphology in static and non-weightbearing position, so we could not investigate the dynamics of the intrinsic foot muscles. Fourth, other intrinsic foot muscles, such as quadratus plantae and extensor digitorum brevis, were not assessed by ultrasonography. Lastly, this study recruited subjects engaged in not the same but a variety of sports (such as baseball, basketball, classic ballet, football, orientating, swimming, tennis, track and field, volleyball, and weightlifting). Therefore, further research including a variety of loading conditions and/or intrinsic foot muscles in the same sports player is needed to understand foot functions during sports-related movement.

## 5. Conclusions

In conclusion, there were significant correlations between the thickness of the plantar fascia and the CSA of FHB and VGRFmax during landing, and correlations between the plantar fascia and AbH and reactivity during repetitive jumping among healthy adults. Thus, the present findings suggest that the CSA of FHB contributes to force attenuation during landing, and that the thickness of the plantar fascia and the CSA of AbH facilitate jumping high with minimal contact time during repetitive rebound movement performed by healthy adults.

## Figures and Tables

**Figure 1 ijerph-18-04511-f001:**
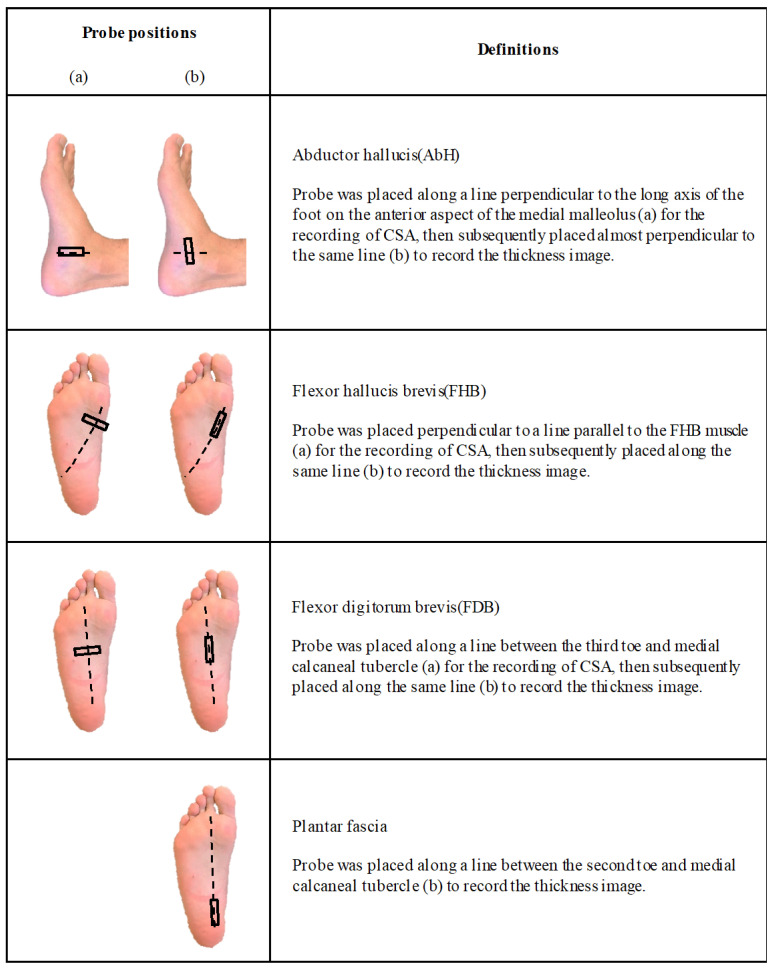
The probe’s position in the assessment of selected tissue thickness and CSA. Abbreviation: AbH, abductor hallucis; FHB, flexor hallucis brevis; FDB, flexor digitorum brevis.

**Figure 2 ijerph-18-04511-f002:**
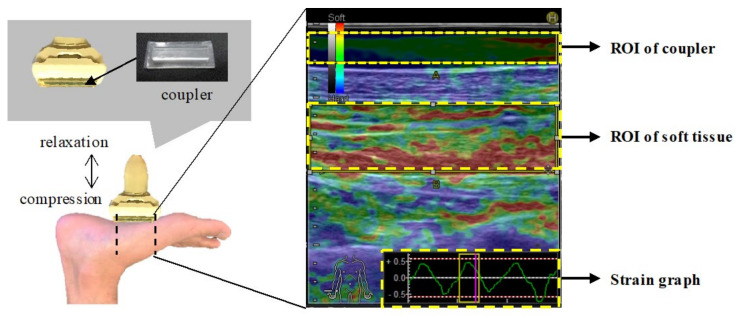
Assessment of muscle hardness using RTE and RTE images. We pressed the transducer against the selected soft tissue with constant repeated pressure (compression–relaxation cycle) by checking the strain graph. Subsequently, we selected the ROI of the coupler and soft tissue, and the strain ratio (muscle/coupler) was calculated. Abbreviation: RTE, real-time tissue elastography; ROI, region of interest.

**Table 1 ijerph-18-04511-t001:** Median (interquartile range) and Mean (standard deviation) Values of Assessed Parameters.

Parameters		Median (Interquartile Range)	Mean (Standard Deviation)
Age	(years)	23.0 (22.0–24.5)	23.3 ± 1.9
Body height	(cm)	169.0 · (164.5–173.0)	169.8 ± 0.8
Body weight	(kg)	60.0 · (54.0–65.5)	59.9 ± 9.0
BMI	(kg/m^2^)	20.3 · (19.0–23.1)	20.7 ± 2.2
Foot length	(cm)	24.7 · (24.1–25.7)	24.8 ± 1.4
Thickness of selected tissues (mm)			
Abductor hallucis		11.0 · (9.5–12.4)	11.1 ± 1.9
Flexor hallucis brevis		12.2 · (9.6–13.5)	11.4 ± 2.4
Flexor digitorum brevis		9.6 · (6.3–11.3)	8.8 ± 2.7
Plantar fascia		3.0 · (2.9–3.2)	3.0 ± 0.4
Cross-sectional area of selected tissues (mm^2^)			
Abductor hallucis		261.3 · (179.0–285.8)	245.0 ± 66.2
Flexor hallucis brevis		236.2 · (220.8–273.7)	244.8 ± 39.0
Flexor digitorum brevis		204.0 · (175.2–259.2)	214.3 ± 50.4
Ultrasound elastography of selected tissues (muscle/coupler)			
Abductor hallucis		1.20 · (0.65–1.95)	1.28 ± 0.76
Flexor hallucis brevis		1.60 · (1.30–2.21)	1.95 ± 1.03
Flexor digitorum brevis		2.60 · (1.70–3.70)	2.70 ± 1.24
Plantar fascia		0.30 · (0.20–0.55)	0.40 ± 0.29
Drop landing test			
COP length	(%foot length)	24.6 · (21.6–27.3)	25.4 ± 4.8
VGRFmax	(%BW)	439.2 · (398.0–540.8)	458.4 ± 90.6
Peak time	(ms)	54.0 · (48.8–62.8)	57.2 ± 10.4
Loading rate	(%BW/s)	7.7 · (6.5–10.1)	8.3 ± 2.5
Repetitive rebound jump test			
Contact time	(s)	0.30 · (0.27–0.33)	0.30 ± 0.05
Jump height	(cm)	12.9 · (10.8–16.3)	13.5 ± 3.5
Reactive jump index	(cm/s)	45.6 · (34.9–53.4)	46.0 ± 15.1

Abbreviation: BMI, body mass index; COP, center of pressure; VGRFmax, maximum vertical grand reaction force; BW, body weight.

**Table 2 ijerph-18-04511-t002:** Correlation Analysis between Ultrasonographic Measurements and Repetitive Rebound Jump and Single-Leg Drop Landing Assessment.

Variables		Drop Landing Test								Repetitive Rebound Jump Test					
		COP Length		VGRF Max		Time to Peak		Loading Rate		Contact Time		Jump Height		Reactive Jump Index	
		*r*	*p*	*r*	*p*	*r*	*p*	*r*	*p*	*r*	*p*	*r*	*p*	*r*	*p*
Thickness of selected tissues	(mm)														
Abductor hallucis ^a^		0.030	0.897	−0.043	0.854	0.093	0.687	−0.059	0.800	−0.230	0.316	0.215	0.349	0.212	0.357
Flexor hallucis brevis ^a^		−0.229	0.319	−0.055	0.811	−0.166	0.472	0.137	0.555	−0.050	0.830	0.253	0.269	0.194	0.399
Flexor digitorum brevis ^b^		−0.311	0.169	−0.211	0.360	0.205	0.373	−0.216	0.346	0.104	0.654	0.360	0.109	0.311	0.169
Plantar fascia of calcaneal portion ^a^		−0.513 *	0.018	0.051	0.825	0.028	0.905	0.083	0.721	−0.425	0.055	0.615 **	0.003	0.645 **	0.002
Cross-sectional area of selected tissues	(mm^2^)														
Abductor hallucis ^a^		−0.055	0.814	−0.051	0.825	0.358	0.111	−0.176	0.447	−0.478 *	0.028	0.258	0.260	0.376	0.093
Flexor hallucis brevis ^a^		−0.163	0.481	−0.472 *	0.031	0.200	0.384	−0.339	0.132	0.117	0.613	−0.066	0.776	−0.059	0.801
Flexor digitorum brevis ^a^		−0.291	0.201	−0.279	0.221	0.278	0.222	−0.351	0.119	0.088	0.706	0.333	0.141	0.240	0.294
Ultrasound elastography of selected tissues	(tissue/ coupler)														
Abductor hallucis ^a^		0.012	0.957	−0.054	0.815	−0.159	0.491	−0.011	0.962	0.159	0.491	0.063	0.786	0.008	0.971
Flexor hallucis brevis ^b^		−0.023	0.920	−0.017	0.942	−0.147	0.526	0.032	0.891	−0.259	0.257	0.190	0.410	0.227	0.322
Flexor digitorum brevis ^a^		0.004	0.987	0.292	0.199	−0.220	0.338	0.267	0.242	−0.194	0.400	0.073	0.753	0.153	0.507
Plantar fascia ^b^		0.219	0.341	−0.071	0.760	−0.150	0.515	−0.028	0.903	−0.092	0.692	−0.299	0.187	−0.211	0.358

* *p* < 0.05, ** *p* < 0.01. ^a^ Each variable was tested by the Pearson correlation coefficient. ^b^ Each variable was tested by the Spearman’s rank correlation coefficient. Abbreviation: COP, center of pressure; VGRFmax, maximum vertical grand reaction force normalized by body weight.

**Table 3 ijerph-18-04511-t003:** Multiple Regression Analysis of the Association between Ultrasound Assessment and Repetitive Rebound Jump.

Variables	Contact Time				Reactive Jump Index			
	β	95% CI interval		*p*	β	95% CI interval		*p*
		Lower	Upper			Lower	Upper	
CSA of abductor hallucis	−0.369	−0.001	0.000	0.101	0.152	−0.057	0.126	0.437
Thickness of plantar facia	−0.284	−0.086	0.019	0.200	0.587	7.219	38.743	0.007

Explanatory variables were CSA of abductor hallucis and thickness of plantar fascia. β, standardized partial regression coefficient; CI, confidential interval; COP, center of pressure; CSA, cross-sectional area.

## Data Availability

The data presented in this study are available on request from the corresponding author. The data are not publicly available due to ethical consideration.

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
