# Peer review of "Contribution of Plantar Fascia and Intrinsic Foot Muscles in a Single-Leg Drop Landing and Repetitive Rebound Jumps: An Ultrasound-Based Study"

_ijerph, 2021, doi:10.3390/ijerph18094511_

Round 1

Reviewer 1 Report

The overall content is scientifically sound, and the results provide new information regarding the use of Ultrasonography as a assessing tool and its relationship with the actual performance. Below are the comments requiring to be addressed.

  1. Page2 line 44. The authors cited ref. 5 to address the contribution of lower-limb stiffness to jump height and ground reaction time at the time of jumping. However, it seems that there is no discussion of the lower-limb stiffness in ref 5 and hence, clarification may be needed.
  2. Page2 line 49. Authors mentioned ref 7 as an example of investigating the relationship between morphologies of the plantar fascia and IFM and gait. However, ref. 7 investigated the physical demands of elite basketball player without mentioning the morphology of muscles. Clarification is needed.
  3. Page2 line 50. Ref. 8 investigated the effect of plantar intrinsic foot muscle strengthening exercise on static and dynamic foot kinematics without using ultrasonography for outcome variables. Clarification is needed.
  4. Page2 line 50. The order of Ref. 9 and 10 should be reversed. Because ref. 9 is about dynamic running and ref. 10 is about single-leg standing.
  5. Page2 line 82. Authors chose abductor hallucis (AbH), flexor hallucis brevis (FHB), and flexor 82 digitorum brevis (FDB) muscles as the representation of intrinsic foot muscles. However, there are other important significant intrinsic muscles such as quadratus plantae as used by reference 3 cited by the authors. Please provide justification and reason why used there three above mentioned muscles in the current study.
  6. Page 5 line 178. Please provide reference or explanation for the reactive jump index (RJI).
  7. Page 6 line 213. The caption for table 2 should be separated from the table 1.

Discussion

  1. In the current study, authors investigated the thickness, cross-sectional area, and hardness of the muscles, however, no significant finding was found regarding the performance and the hardness of the muscles. Please provide discussion regarding the possible reason why the hardness didn’t contribute the performance.
  2. Although the use of Ultrasonography may provide the morphology of the muscles, the muscles were measured during static and non-weightbearing position. Would it be considered as a limitation for the current study?
  3. The results showed the significant correlation between thickness of plantar fascia and COP length, but related discussion is lacking. Please provide the possible mechanism accounting for such relationship.
  4. Page 8 line 245. Since big portion of the findings of the current study is related to the RJI, the detailed description regarding the significance of RJI should be provided.

Author Response

[10th, April, 2021]

Dear Reviewer #1,

I wish to resubmit the original article for publication in the International Journal of Environmental Research and Public Health, titled “Plantar fascia and intrinsic foot muscles in a single leg drop and repetitive rebound jumps: An ultrasound-based study.” The manuscript ID is ijerph-1177890. The manuscript has been rechecked and appropriate changes have been made in accordance with the reviewers’ suggestions. The responses to the comments have been prepared and given below.

We thank you and the reviewers for your thoughtful suggestions and insights, which have enriched the manuscript and produced a better and more balanced account of the research. We hope that the revised manuscript is now suitable for publication in your journal.

This manuscript has not been published or presented elsewhere in part or in entirety and is not under consideration by another journal. All study participants provided informed consent, and the study design was approved by the appropriate ethics review board. We have read and understood your journal’s policies, and we believe that neither the manuscript nor the study violates any of these. There are no conflicts of interest to declare.

Thank you for your consideration. I look forward to hearing from you.

Reviewer 2 Report

The work entitled " Contribution of plantar fascia and intrinsic foot muscles in a single-leg drop landing and repetitive rebound jumps: An ultrasound-based study " is a very interesting job. Therefore, I consider that it has an interesting approach for publication in Int. J. Environ. Res. But, there are some questions of form that should be taken into account prior to consider this article for publication.

I enclose my suggestions for consideration by the authors.

  • In the introduction, the authors describe that the foot is made up of 26 bones… But… Are they 26 or 28?
  • What is the objective of making these measurements?, It is not clear to me
  • In the methodology, the authors do not reflect where the subjects were recruited.
  • The manuscript does not reflect whether the subjects studied perform the same sporting activity, this is an interesting fact that must be discussed ... the response of the fascia can behave differently when the subject practices different sports?
  • Sections 2.4 and 2.5, these are difficult to follow an easy comprehensive reading. Recommending the authors to formulate these sections more concisely and clearly.
  • It would be interesting for the authors to state in the discussion what benefits their results bring to sports practice.
  • The conclusions do not respond clearly to the objective.
  • In some references the document pages are missing.

Author Response

[10th, April, 2021]
Dear Reviewer #2,

I wish to resubmit the original article for publication in the International Journal of Environmental Research and Public Health, titled “Plantar fascia and intrinsic foot muscles in a single leg drop and repetitive rebound jumps: An ultrasound-based study.” The manuscript ID is ijerph-1177890. The manuscript has been rechecked and appropriate changes have been made in accordance with the reviewers’ suggestions. The responses to the comments have been prepared and given below.

We thank you and the reviewers for your thoughtful suggestions and insights, which have enriched the manuscript and produced a better and more balanced account of the research. We hope that the revised manuscript is now suitable for publication in your journal.
This manuscript has not been published or presented elsewhere in part or in entirety and is not under consideration by another journal. All study participants provided informed consent, and the study design was approved by the appropriate ethics review board. We have read and understood your journal’s policies, and we believe that neither the manuscript nor the study violates any of these. There are no conflicts of interest to declare.

Thank you for your consideration. I look forward to hearing from you.
